

# The use of audio self-hypnosis to promote weight loss using the transtheoretical model of change: a randomized clinical pilot trial

Jumana Antoun[1], Marielle El Zouki[1] and Myrna Saadeh[2]

[1] Family Medicine, American University of Beirut, Beirut, Lebanon
[2] Wellbeing Center, Beirut, Lebanon

## ABSTRACT

**Background:** Few studies suggest the use of hypnosis in weight loss may be beneficial, especially when in conjunction with other lifestyle modifications or cognitive behavioral therapy. The primary aim of this study was to determine the ability of self-hypnosis audiotape to promote weight loss by measuring its effects on the Transtheoretical Model (TTM) of change stages and processes.

**Methods:** This study is a 3-week randomized double-blinded parallel controlled trial among adults who have overweight or obesity. The intervention group listened to a self-hypnosis audio file while the control group listened to a placebo audio file.

**Results:** Forty-six participants completed the 3-week follow-up visit. There was no association between progression across stages of change and self-hypnosis (X2(2, 46) = 1.909, $p$-value = 0.580). Gender, baseline BMD, and baseline S-weight had no effect on the association between stage change progression and self-hypnosis. The mean difference in weight at 3 weeks was −0.63 ± 0.43 kg in the hypnosis group and 0.0 ± 1.5 kg in the control group, independent t-test, $p$ = 0.148.

**Conclusion:** Self-hypnosis was not associated with a progression in the TTM's stages of change or with weight loss after 3 weeks. As this pilot study was underpowered, further research with larger sample size and an examination of the effect of various self-hypnosis content and duration is recommended.

## INTRODUCTION

Obesity has recently emerged as a major health concern in high-income countries and low- and middle-income countries, particularly in urban areas. Obesity and overweight are linked to increased mortality and morbidity, accounting for 4.0 million deaths and 120 million disability adjusted life years (DALYs) globally (*Collaborators, 2017*). Diet, exercise, and behavioral modification are the foundation of all obesity management approaches, with pharmacotherapy and bariatric surgery used as adjuncts based on BMI cutoffs and the presence of comorbidities (*Semlitsch et al., 2019*). Few studies have investigated the potential use of complementary and alternative therapies for weight loss

Corresponding author
Jumana Antoun, ja46@aub.edu.lb

(*Esteghamati et al., 2015*), with some evidence indicating that hypnosis may be effective in treating obesity (*Entwistle et al., 2014*; *Milling, Gover & Moriarty, 2018*).

Hypnosis is defined as "a state of consciousness involving focused attention and reduced peripheral awareness characterized by an enhanced capacity for response to suggestion" (*Elkins et al., 2015*). In clinical practice, hypnosis is used to treat various physical and behavioral conditions, with growing evidence demonstrating its efficacy as an adjuvant intervention in psychotherapy and health care (*Adachi et al., 2014*; *Kirsch, Montgomery & Sapirstein, 1995*; *Schaefert et al., 2014*; *Terhune et al., 2017*). The application of hypnosis consists primarily of an induction phase to focus the patient's attention, followed by recommendations to induce relaxation and thus a hypnotic state (*i.e.*, an altered state of consciousness) in which the person may be more receptive to positive feedback directions. There are numerous techniques for inducing a hypnotic trance, the most common being communication, particularly verbal communication (*Faghfoory, 2018*).

When it comes to obesity and weight loss, hypnosis has been used either independently or in conjunction with other methods of treating obesity (*Entwistle et al., 2014*; *Esteghamati et al., 2015*; *Milling, Gover & Moriarty, 2018*; *Ramondo et al., 2021*). Although previous research has demonstrated a beneficial effect of hypnosis on weight loss (*Barabasz & Spiegel, 1989*; *Bolocofsky, Spinler & Coulthard-Morris, 1985*; *Cochrane & Friesen, 1986*; *Davis & Dawson, 1980*; *Green, 1999*), they may be criticized for their varied methodologies and use of various hypnotic methods, including self-hypnosis, hypnotic audiotapes, or specialized hypnotists. The sample size ranged from case studies (*Davis & Dawson, 1980*; *Green, 1999*) to more than 100 participants (*Bolocofsky, Spinler & Coulthard-Morris, 1985*). On the other hand, a recent randomized controlled trial demonstrated that while self-hypnosis did not result in statistically significant weight loss, it did result in increased satiety and quality of life (*Bo et al., 2018*). Hypnosis is considered a promising treatment option for obesity when used with cognitive-behavioral hypnotherapy techniques (*Milling, Gover & Moriarty, 2018*; *Ramondo et al., 2021*).

The effects of hypnosis may not be directly related to weight loss but to behavior modification. Audiotape hypnosis was effective in facilitating progression through stages of change towards smoking cessation and a reduction in daily cigarette consumption (*Munson, Barabasz & Barabasz, 2018*). The transtheoretical model (TTM) processes and stages of change are predictors of behavior change in interventions advocating healthy behaviors (*Andres, Saldana & Gomez-Benito, 2009*). The TTM classifies behavior change into five stages: pre-contemplation, contemplation, preparation, action, and maintenance. TTM can be used to guide the assessment of the patient's readiness and the selection of the most appropriate and effective technique or intervention for promoting dietary changes and exercise (*de Freitas et al., 2020*). Therefore, this study aims to assess the effect of audiotaped self-hypnosis on the stages and processes of change as defined by the TTM of change using a randomized controlled design. A secondary outcome was the effect of audiotaped self-hypnosis on weight loss.

## MATERIALS AND METHODS

This was a 3-week double-blinded, randomized, parallel, placebo-controlled trial. Both audio files, which were distributed to participants *via* USB, lasted 15 min and were prepared in English by the same experienced hypnotist. The control audio file started with an induction phase followed by direct messages to change one's lifestyle, from eating habits to exercise. The self-hypnosis audio file included the same direct messages embedded within the stages of hypnosis that started with induction, deepening, suggestions, posthypnotic suggestions and alerting. Appendix A contains the script for both the hypnosis and control audio files. Ethical approval was obtained from the Institution Review Board at the American University of Beirut (SBS-2019-0220).

### Recruitment

Recruitment took place *via* (a) emails sent to a random sample of the American University of Beirut and the American University of Beirut Medical Center students and staff and (b) flyers distributed in the Family Medicine clinic's waiting areas. Recruitment occurred between 23/12/2019 and 28/2/2020. Individuals interested in participating in the study could contact the research personnel *via* the contact information (phone number and email address) included in the invitation emails and flyers. Following that, a 5-min phone call was made to all interested participants to determine their preliminary eligibility.

### Participants

The inclusion criteria were as follows: BMI 25 kg/m$^2$ or greater, aged 18 to 64 years, capable of providing written informed consent, fluent in English, all men, women who are not planning to become pregnant, individuals who have attempted to lose weight previously, were planning to lose weight within the next 6 months or were dissatisfied with the results of their current weight loss plan. The exclusion criteria were as follows: illiterate, individuals diagnosed with a psychotic disorder, who are currently on antipsychotic medication, pregnant women, women planning to become pregnant during the study period, and those who did not intend to lose weight within the next 6 months or are satisfied with their weight loss progress.

### Measurements

S- and P-Weight were used to assess both processes and the stages of change following the TTM (*Andres, Saldana & Gomez-Benito, 2011*). The S- and P-Weight are two self-report questionnaires, respectively investigating the stages of change and the processes of change (*Prochaska, 1985*). The S-Weight comprises five mutually exclusive items that correspond to the five stages of change: Precontemplation, Contemplation, Preparation, Action, and Maintenance. The P-Weight is a 34-item questionnaire that assesses an individual's readiness to engage in a diet and physical activity. The four processes of change measured by the P-Weight are Emotional Re-evaluation (EmR), Weight Management Actions (WMA), Environmental Restructuring (EnR), and Weight Consequences Evaluation (WCE).

The PHQ-2 questionnaire consists of two questions that assess the frequency of depressive and anhedonia symptoms. Each question was scored from 0 (not at all) to 3 (nearly every day). PHQ-2 has a sensitivity of 0.86 and a specificity of 0.78 to screen for depression at a cutoff point of 3 or more (*Arroll et al., 2010*).

## Study protocol

Participants were asked to attend a baseline 1-h visit and 20-min follow-up visit at 3 weeks. The research assistant rescreened participants for inclusion and exclusion criteria during the baseline visit. Female participants underwent a pregnancy test. The research assistant took anthropometric measurements (weight, height, and waist circumference). Weight and height were measured on the same scale (DetectoScale), without shoes and wearing indoor clothing. Waist circumference was measured starting at the top of the hip bone, wrapping circumferentially around the body, and leveling with the belly button in a private room. Following the assessment, each participant completed a Kobo survey in a private room using their assigned code targeting sociodemographic information about their age, gender, level of education, employment status, and previous and current weight loss methods. The PHQ2 questionnaire followed this to screen for depression and the S- and P-weight tools to collect the pre-intervention stage of change. The participants were then randomly assigned to one of two study groups and were asked to listen to the assigned audio file at the baseline visit. Block randomization was generated using the randomization.com website, (http://www.randomization.com). An independent person prepared opaque envelopes for the USBs and labeled them with serial numbers. In a separate data file, serial numbers were linked to the actual audio file (hypnosis or placebo) and were revealed after the data was collected. The participants and research assistants were blinded to which group the participant was allocated to. Each participant received a USB flash drive containing the assigned audio file and was instructed to listen to it nightly for at least 7 days using headphones in a private room and as needed for reinforcement after that. By the end of the visit, participants had received pamphlets containing information on dietary modification for weight loss. The participants were informed that hypnosis will be used in the intervention group. They were given a brief education about hypnosis, informing the participants that they can stop the recording anytime and that they are free to accept or reject any suggestions given to them. During the follow-up visit, the research assistant repeated anthropometric measurements, and administered the S- and P-weight questionnaires and a questionnaire that assessed the use of the audio file and any change in lifestyle towardweight loss. Women were asked about their pregnancy status during the follow-up visits.

## Sample size calculation

The sample size was calculated with the primary outcome defined as a step-up transition in at least one stage of change as defined by the S-Weight tool. Available data on dietary counseling and physical activity in general practice showed that approximately 10–15% of participants progressed on the contemplation and action scales as defined by the transtheoretical model without any intervention (*Munson, Barabasz & Barabasz, 2018*).
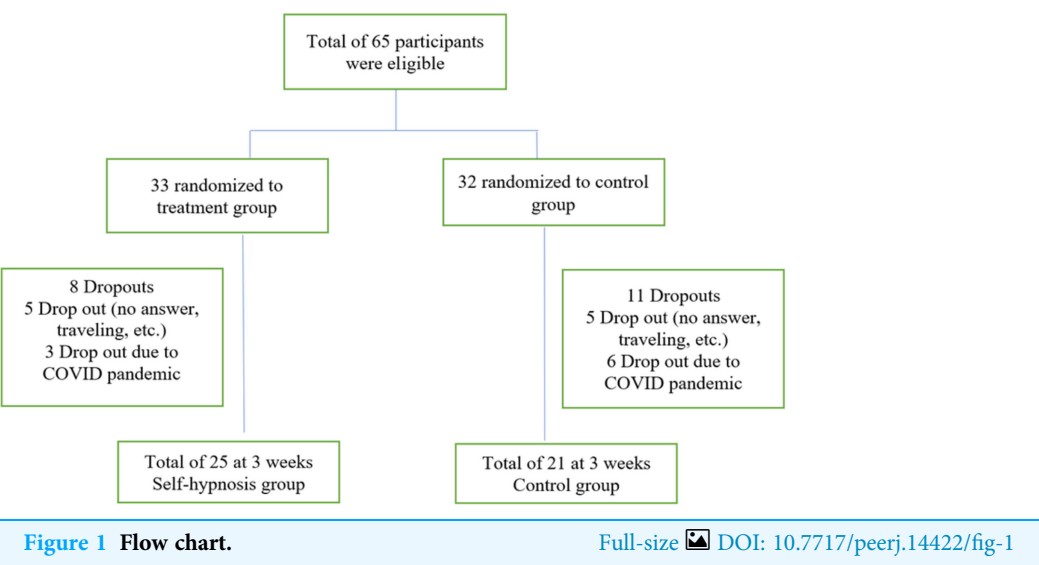

**Figure 1  Flow chart.**                               

Participants who received an additional hypnotic intervention had an additional 15–20% progression of the stages, summing up to 30–35%. Therefore, we assumed that 10% of participants in the placebo group and 30% in the self-hypnosis would change at least one stage. The *p*-values were set at 0.05 and power at 80%, while accounting for 20% attrition, the sample size calculation for each group is 80 using G*Power 3.1.9.7 software.

## Statistical analysis

Descriptive statistics were used for the demographics. The primary outcome measure was a categorical variable with three options: proportion of participants who had negative progression of at least one stage, no change in the stages, and positive progression of at least one stage. Secondary outcome measures were the difference in weight, waist circumference, and P Weight processes. Chi-squared was used to determine the association between the two groups and the stages of change. An independent t-test was used to determine the association between the two groups and weight, waist circumference, and P-weight. *p*-values were set at 0.05. Data were analyzed using IBM SPSS Statistics (Version 27).

## RESULTS

Sixty-five participants met the eligibility criteria and completed the initial visit. Nineteen participants dropped out at week three, leaving 46 participants who completed the initial and 3-week follow-up visit (Fig. 1). The two groups shared similar demographic characteristics (Table 1). Most respondents were between 30 and 59 years, females, and had a graduate degree. Their BMI was in the obesity range. Fewer than a quarter reported high scores on PHQ2 indicative of depression. One-third of respondents were in the contemplation stage, while nearly half were in the action stage. During the first week, half of the participants (*n* = 20, 43.5%) listened to the audiofile at least once daily. During the 3 weeks, the participants listened to the audiofile at home at least once daily (*n* = 15,

**Table 1 Demographics (N = 65)\*.**

| | Intervention group (N = 33) n (%) | Control group (N = 32) n (%) | p-value |
|---|---|---|---|
| **Age** | | | 0.432 |
| 18–20 | 2 (6.1) | 4 (12.9) | |
| 21–29 | 5 (15.2) | 8 (25.8) | |
| 30–39 | 14 (42.4) | 7 (22.6) | |
| 40–49 | 8 (24.2) | 7 (22.6) | |
| 50–59 | 4 (12.1) | 5 (16.1) | |
| **Gender** | | | 0.166 |
| Male | 6 (18.2) | 11 (34.4) | |
| Female | 27 (81.8) | 21 (65.6) | |
| **Educational level** | | | 0.594 |
| High school degree or less/Technical degree | 5 (15.1) | 4 (12.5) | |
| Undergraduate degree | 8 (25.0) | 8 (25.0) | |
| Graduate degree | 14 (43.8) | 18 (56.3) | |
| Post-Graduate degree | 5 (15.6) | 2 (6.3) | |
| **Employment status** | | | 0.600 |
| Full time job | 24 (75.0) | 25 (78.1) | |
| Student | 8 (25.0) | 7 (21.9) | |
| **BMI (kg/m$^2$)** | | | 0.400 |
| 25–29.9 | 11 (52.4%) | 7 (36.8%) | |
| 30–34.9 | 6 (28.9%) | 9 (47.4%) | |
| >=35 | 4 (19.0%) | 3 (15.8%) | |
| **Exercise frequency** | | | 0.635 |
| Once a week | 7 (21.2) | 5 (15.6) | |
| 2–3 times a week | 8 (24.2) | 11 (34.4) | |
| Almost everyday | 18 (54.5) | 16 (50.0) | |
| **Obesity duration** | | | 0.683 |
| During childhood | 6 (18.2) | 8 (25.0) | |
| Since puberty | 5 (15.2) | 6 (18.8) | |
| During adulthood | 22 (66.7) | 18 (56.3) | |
| **Weight loss plan in the past year** | | | 0.995 |
| Never | 4 (12.1) | 4 (12.5) | |
| 1–2 times | 20 (60.6) | 19 (59.4) | |
| More than twice | 9 (27.3) | 9 (28.1) | |
| **Junk food consumption** | | | 0.718 |
| Once a week | 19 (30.3) | 7 (22.6) | |
| 2–3 times a week | 13 (39.4) | 15 (48.4) | |
| Almost everyday | 10 (30.3) | 9 (29.0) | |
| **PHQ2 score** | | | 0.213 |
| Less than 3 | 23 (74.2) | 28 (87.5) | |
| 3 or more | 8 (25.8) | 4 (12.5) | |

|  | Intervention group (N = 33) n (%) | Control group (N = 32) n (%) | p-value |
|---|---|---|---|
| **Motivation for weight loss (more than one answer allowed)** |  |  |  |
| 1- Improve health | 26 (78.8) | 30 (93.8) | 0.081 |
| 2- Feel more attractive | 20 (60.6) | 15 (46.9) | 0.267 |
| 3- Please others | 10 (30.3) | 9 (28.1) | 0.847 |
| 4- Have more confidence | 20 (60.6) | 15 (46.9) | 0.267 |
| 5- Fit my favorite clothes | 24 (72.7) | 19 (59.4) | 0.255 |
| 6- Ease joint pains | 12 (36.4) | 8 (25.0) | 0.321 |
| **S-Weight** |  |  | 0.714 |
| At the moment, I'm not doing anything to lose weight, and I have no intention of doing anything over the next 6 months | (Exclusion criteria) | (Exclusion criteria) |  |
| At the moment, I'm not doing anything to lose weight, but I'm thinking about doing something over the next 6 months | 11 (33.3) | 10 (33.3) |  |
| During the last year, I haven't done anything to lose weight but I'm planning to do something over the next 30 days | 6 (18.2) | 3 (9.4) |  |
| I've been making an effort to lose weight (by dieting and/or exercising) in the past 6 months | 14 (42.4) | 16 (50.0) |  |
| I've been making an effort to maintain my weight (by dieting and/or exercising) for more than 6 months | 2 (6.1) | 3 (9.4) |  |
|  | N = 25 | N = 21 |  |
| **Frequency of listening to the audio in the 3 weeks** |  |  | 0.339 |
| At least once daily | 8 (32.0) | 7 (33.3) |  |
| At least once a week | 6 (24.0) | 6 (28.6) |  |
| 1–3 times | 4 (16.0) | 0 (0) |  |
| As I felt needed, but I can not remember | 2 (8.0) | 4 (19.0) |  |
| Never | 5 (20.0) | 4 (19.0) |  |
| **Frequency of listening to the audio in the first week** |  |  |  |
| More than once daily | 1 (4.0) | 3 (14.3) | 0.079 |
| At least once daily | 8 (32.0) | 8 (38.1) |  |
| 3–5 times | 9 (36.0) | 2 (9.5) |  |
| 1–2 times | 5 (20.0) | 4 (19.0) |  |
| Never | 1 (4.0) | 0 |  |

**Note:**
*Missing values exist.

32.6%) and 20% ($n = 9$) have never listened to the audio beyond the recruitment baseline visit.

There was no association between self-hypnosis and progression across stages of change ($X2(2, 46) = 1.909$, $p$-value = 0.580) (Table 2). Gender, baseline BMD, and baseline S-weight had no effect on the association between stage changes progression and self-hypnosis.

The mean difference in weight at 3 weeks was $-0.63 \pm 0.43$ Kg in the hypnosis group and $0.0 \pm 1.5$ kg in the control group, (independent t-test, $p = 148$). The waist circumference changed by $-1.2 \pm 3.5$ cm in the hypnosis group and by $-0.72 \pm 3.7$ cm in

**Table 2 The difference in the stages of change in 3 weeks (N = 46).**

|  | Negative progression of at least one stage n (%) | No change in the stage n (%) | Positive progression of at least one stage n (%) | p-value (Chi-square) |
|---|---|---|---|---|
| Self-hypnosis | 7 (28.0) | 13 (52.0) | 5 (20.0) | 0.580 |
| Control | 3 (14.3) | 15 (71.4) | 3 (14.3) |  |

the control group, leading to a −0.48 ± 1.1 change in waist circumference favoring self-hypnosis (independent t-test, $p = 0.653$). Participants in the hypnosis group were more likely to have followed a weight loss plan (48.0%) than those in the control group (33.3%), but the difference is not statistically significant (Fisher's exact test, $p = 0.377$). There was no statistically significant difference in P weight items between the two groups between baseline and 3 weeks.

## DISCUSSION

The primary objective of this randomized, double-controlled trial was to determine the effect of audio self-hypnosis on the stages of change, with the primary outcome defined as a progression in at least one stage. This pilot study did not show an effect of audio self-hypnosis on the TTM's stages of change or weight loss at 3 weeks. A narrative review of the effects of hypnosis on weight reduction revealed that most studies included a lifestyle modification component in addition to the hypnosis (*Roslim et al., 2021*). This study aimed to use self-hypnosis as the main intervention because it is a scalable, noninvasive, and inexpensive intervention. The primary outcome was the stage of change progression, not weight loss. The stage of change correlates positively with motivation (*Keating & McCurry, 2019*), which results in successful weight loss. The frequency with which hypnosis is used is related to weight changes (*Bo et al., 2018*). Only 43.5% of participants listened to the audio file consistently daily. Further research is needed to explore the effect of self-hypnosis with varying content and duration on stages of change and weight loss.

This study found that audio self-hypnosis had no statistically significant effect on weight loss after 3 weeks. Similarly, self-hypnosis resulted in a similar weight reduction as diabetic education in patients with diabetes type II. (*Levenson & David Levenson, 2018*). Self-hypnosis was associated with increased satiety and quality of life in patients with obesity but not with weight loss (*Bo et al., 2018*). When self-hypnosis was used as part of the cognitive-behavioral intervention in addition to a hypnosis intervention, the effect size on weight loss was larger than when self-hypnosis was not used (*Milling, Gover & Moriarty, 2018*). The effect of self-hypnosis on weight loss is still scarce and controversial, particularly when used alone.

### Strength and limitations

Most of the research on hypnosis and self-hypnosis is observational and was published two decades ago (*Milling, Gover & Moriarty, 2018*; *Pellegrini et al., 2021*; *Roslim et al., 2021*). This was a double-blinded study in which self-hypnosis was used as the sole intervention.

A certified hypnotist created the hypnosis audio files. We did not ask the participants to guess their allocation. The study's low sample size is one of its most significant limitations. We encourage the reader to accept the results cautiously as the study is underpowered. Based on the actual achieved sample size, the achieved power is 49%. Further research is needed to include a larger sample size. The COVID pandemic has influenced recruitment and retention rates. The majority of the sample presented in this study was recruited between Dec 2019 and Feb 2020, before the first case of COVID was declared in the country (Feb 21, 2020). The first lockdown in the country occurred in May 2020. Similar to other studies regarding hypnosis (*Pellegrini et al., 2021*), dropout and recruitment are major obstacles. Another limitation is the sample's limited generalizability due to its predominantly educated female participants. Lastly, this study explored audio self-hypnosis and does not generalize to other forms of hypnosis.

## CONCLUSIONS

This pilot study did not show an effect of audio self-hypnosis on the TTM's stages of change or weight loss at 3 weeks. Further research with a larger sample size and an examination of the effect of various self-hypnosis content and duration is recommended.

### Funding
The authors received no funding for this work.

### Competing Interests
Jumana Antoun is an Academic Editor for PeerJ. Myrna Saadeh is the founder of the WellBeing Center.

### Author Contributions
- Jumana Antoun conceived and designed the experiments, analyzed the data, prepared figures and/or tables, authored or reviewed drafts of the article, and approved the final draft.
- Marielle El Zouki conceived and designed the experiments, performed the experiments, prepared figures and/or tables, authored or reviewed drafts of the article, and approved the final draft.
- Myrna Saadeh conceived and designed the experiments, authored or reviewed drafts of the article, and approved the final draft.

### Human Ethics
The following information was supplied relating to ethical approvals (*i.e.*, approving body and any reference numbers):

The American University of Beirut Institution Review Board granted ethical approval to carry out the study within its facilities. (Ethical Application Ref: SBS-2019-0220).

## Clinical Trial Ethics

The following information was supplied relating to ethical approvals (*i.e.*, approving body and any reference numbers):

The American University of Beirut Institution Review Board granted ethical approval to carry out the study within its facilities. (Ethical Application Ref: SBS-2019-0220).

## Data Availability

The raw data are available in the Supplemental Files.

## Clinical Trial Registration

The following information was supplied regarding Clinical Trial registration:

NCT04247568.

## Supplemental Information

Supplemental information for this article can be found online at http://dx.doi.org/10.7717/peerj.14422#supplemental-information.

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
