# Peer review of "The use of audio self-hypnosis to promote weight loss using the transtheoretical model of change: a randomized clinical pilot trial"

_PeerJ, doi:10.7717/peerj.14422_

## Round 0.1 · original submission · Major Revisions

I would like to encourage you to revise and resubmit. I think your manuscript is well written and the study is very interesting, and seems well designed. As you will see, both reviewers were worried about the final sample size and felt you did not deal with it in a satisfactory fashion. Both reviewers made specific recommendations on how to do so. I suggest you go through the comments of both reviewers systematically and address them.

·

Basic reporting

This article is well written. The literature references and background provided are sufficient .

Experimental design

The research question well defined, relevant & meaningful. The experimental design is sound. There are two problems with the design. the first is the small sample size. the second is that the experiment was done during COVID pandemic. See comment 4

Validity of the findings

The small sample size and COVID pandemic damage the validity of the findings.

Additional comments

This is a randomized control study of weight loss using audio self-hypnosis.
I enjoyed reading this ms. It is well-written and the idea is interesting.
Although the results are not statistically significant, journals should encourage authors to report non-significant results, as long as the methodology is sound.
My main concern is for the sample size. There really aren’t enough participants to assess so many statistical analyses. I would expect that with a greater sample size, the authors could achieve more informative results. Also, I see that part of the study was assessed during COVID pandemic. We know from research that many people changed their dieting habits due to lockdowns. I would recommend the project be assessed again without the pandemic’s influence on a larger sample size.
This study could be used as a pilot study for future research. Maybe the authors should add to the title – pilot study.

Abstract:
1. The abstract is well-written
Introduction:
2. The Introduction is well written.

Methods:
3. This section is well written.
4. My main concern is for the sample size. There really aren’t enough participants to assess so many statistical analyses. I would expect that with a greater sample size, the authors could achieve more informative results. Also, I see that part of the study was assessed during COVID pandemic. We know from research that many people changed their dieting habits due to lockdowns. I would recommend the project be assessed again without the pandemic’s influence on a larger sample size.

Results:
5. The results are well-written.

Discussion:
6. The discussion is well-written.

·

Basic reporting

The paper describes a randomized, double blind, prospective controlled study of self-hypnosis via audio recording to facilitate weight loss. The paper is well written and it is of interest to the readership. I am happy that the authors provide the raw data and also script of the intervention audio (but this was not attached to the review materials). My main concern related to the manuscript is that the study is described as a confirmatory study, but the recruitment was not successful, and the originally set 160 participant target was not reached (only 46 participants were included in the end). It would be important to be explicit about this fact in the sample size justification of the paper, the results section, and also the discussion, because the low power precludes the interpretation of the null results in this investigation. Here are some specific comments that might help to improve the quality of the manuscript:
- It should be clearly stated in the manuscript, what inforation was given to the participants about the intervention. Specifically: Was it made clear for them that they will receive hypnosis (if they are allocated to the intervention group)? Was it made clear for them that there is a placebo control group?
- Was there formal hypnosis induction used in the hypnosis audio?
- Delete „The intervention group” in this sentence: „The intervention group Both audio files, which were distributed to participants via USB, lasted 15 minutes and were prepared in English by the same experienced hypnotist.”
- I was happy to read that the script of the hypnosis and control tapes are available to the reader, but unfortunately I was not able to find Appendix A. Please, make sure that this is available with the submission.
- Relatedly, please give some description of the differences in the content of the intervention and the control audios.

Experimental design

- The authors say that „The participants and research assistants were blinded to the intervention”. I don’t think this is what the authors want to say. Probably they intended to say that the participants and research assistants were blinded to which group the participant was allocated to.
- Was maintenance of blinding tested? If so how? For exaple were participants asked to guess which group they were in? Were assistants asked to guess which group participants were in? If so, report data of these checks. If not, talk about this in the limitation or discussion sections.
- The authors descibe a sample size estimation method. There are multiple issues with the described method. First of all, the number calculated does not match the sample size target registered on ClinicalTrials.gov. On the trial registry the authors originally registered a sample size target of 160, while in their calculation in the manuscript they say 72 participants are needed. Another importnat issues is that the effect sizes mentioned from the previous study (10-15% in the control group vs. 30-35% in the treatment group) does not match the numbers used for the power calculation, because the authors claim to have used 10% vs. 45% in Gpower. I am not sure where does 45% come from. It would be more appropriate to use 12.5% vs. 32.5%, the midppoints of the reported prior effect size, or even 15% vs. 30% to be on the safe side. (If the goal is to do a confirmatory study). If I enter 12.5% vs. 32.5% in Gpower (Z-tests > Proportions: Difference between two independent proportions, p2 = 0.325, p1 = 0.125, alpha = 0.05, power = 0.8, allocation ration = 1), I get a target sample size of 106 (which increases to 177 when accounting for 40% dropout rate). This number is much more realistic to have a decent chance for detecting an effect, than the 72 reported in the manuscript, and it is close to the originally registered sample size target on ClinicalTrials.gov.
The authors should be transparent about their process for selecting the target sample size at the registration of the trial. I also would like to suggest to the authors to clearly state in the manuscript that the study is underpowered, and has a low chance of detecting a significant effect for the intervention. When calculating „achieved power” with Gpower, using the same settings as above but using 23 participants per group (the actual achieved sample size), the computed achieved power is 49%, so it is slightly more likely than not that the effect would be undetected by this design.

Validity of the findings

- „The mean difference in weight at three weeks was -0.63±0.43 Kg in the hypnosis group and 0.0±1.5 Kg in the control group, resulting in -0.63±0.43 kg between the hypnosis and the placebo group (independent t-test, p= 148).” -0.63±0.43 kg appears twice in this sentence, this might be a copy-paste error, please, double check.
- Was the amount of home practice (number of times listening ti the audio) assessed somehow? This is hinted at in the Discussion section here: „Only 40% of participants listened to the audio file consistently daily.” If so, please, specify the method via which this was assessed, and report the data in the results section.
- The authors write „waist circumference decreased by -1.2±3.5 cm”. This is a bit confusing: if decrease is negative, it could mean that waist circumferance actually increased. Please, clarify, for example by using the word „changed” instead of „decreased”.
- The aurhors also report that „Participants in the hypnosis group were more likely to have followed a weight loss plan (48.0%) than those in the control group (33.3%) (Fisher Exact Test, p= 0.377).” Since the p-value is not below 0.05, there was no significant difference between the groups. This should be clarified in this sentence, for example by saying „not more likely”, or saying „but this difference was not statistically significant”.
- In the discussion section the authors write that the intervention had no effect. For example: „The study findings indicate that audio self-hypnosis does not affect the TTM stages of change or weight loss at three weeks.” I think all statements declearing no effect for the intervention should be toned down due to the study being underpowered. For example: „our study did not reveal and effect ...”
- This statement is not completely accurate: „This study aimed to use self hypnosis as the sole intervention because it is a scalable, noninvasive, and inexpensive intervention.” According to the description of methods participants did receive a pamphlet about diet for weight loss. This could be clarified.

Additional comments

- I really appreciate that the authors shared raw data. To make this more useful, please, provide data dictionary, which describes what each variable name means, and the meaning of factor levels.
- Also, please, share the analysis code or syntax that reproduces the numbers reported in the manuscript.

---

## Round 0.2 · Minor Revisions

Dear Dr. Antoun:

Your revision answered most of the reviewers' concerns. Well done. However, Reviewer 2, found some minor points to correct, and correcting them will further improve your paper. Please make these additional changes and resubmit.

·

Basic reporting

I feel that the authors have addressed my comments on their ms.

Experimental design

I feel that the authors have addressed my comments on their ms.

Validity of the findings

I feel that the authors have addressed my comments on their ms.

Additional comments

I feel that the authors have addressed my comments on their ms.

·

Basic reporting

I am grateful for the authors for responding to my comments and questions! My comments were largely addressed. However, there remains some issues that I think could still improve the quality of the manuscript. See these comments below:

- I noticed that the last sentence of the conclusion section in the main manuscript (not in the abstract) is incomplete. “…is recommended” or “…is warranted” is missing from the end of the sentence.
- The authors should add an explicit statement in the methods section that indicates that people were informed that hypnosis is going to be used in this study. I now see this in the inform consent form attached as a supplement, but most readers will not have time to check this, so it is important to be in the main text as well.

Experimental design

-

Validity of the findings

- I saw that the authors made some revisions to the limitations section of the work to indicate that the study is underpowered. This is good, but I think that for transparency, it would be also important to change the abstract as well based on this. For example, it could be included in the Conclusion section of the abstract that “…Self-hypnosis was not associated with a progression in the TTM’s stages of change or with weight loss after three weeks. However, the due to low statistical power the generalizability of this finding is unclear…”
- The authors say that since the raw data is present, the results can be easily reproduced. I would beg to differ. It would take many hours for a researcher to reproduce the results in the paper, even if the have the statistical know-how. For checking analytical reproducibility, it would be important that the authors include the SPSS syntax file (not the output file), that would reproduce the numbers reported in the paper.

Additional comments

- When reading the script I found that there are some suggestions that may not be appropriate for everyone, and may even be harmful. For example this one: “You will eat only when you have physiological needs for food and no other time, for the rest of your life.” If someone takes this suggestion literally (and some people do), this can be a potentially harmful suggestion. There are some religious and social customs as well as medical procedures that require a person to eat, even if they do not have physiological need for food. Maybe it would be good to add a sentence when the script is introduced, saying that before reuse, a revision of the script maybe necessary to adopt for local cultural and dietary context.

---

## Round 0.3 · accepted · Accept

Thank you for submitting this paper to PeerJ. I think it is very interesting and look forward to reading more of your work.